# Overview of the Role of Vanillin in Neurodegenerative Diseases and Neuropathophysiological Conditions

**DOI:** 10.3390/ijms24031817

**Published:** 2023-01-17

**Authors:** Clara Iannuzzi, Maria Liccardo, Ivana Sirangelo

**Affiliations:** Department of Precision Medicine, Università degli Studi della Campania “Luigi Vanvitelli”, Via L. De Crecchio 7, 80138 Naples, Italy

**Keywords:** neurological disorders, neurodegeneration, vanillin, natural antioxidants

## Abstract

Nowadays, bioactive natural products play key roles in drug development due to their safety profile and strong antioxidant power. Vanillin is a natural phenolic compound found in several vanilla beans and widely used for food, cosmetic, and pharmaceutical products. Besides its industrial applications, vanillin possesses several beneficial effects for human health, such as antioxidant activity in addition to anti-inflammatory, anti-mutagenic, anti-metastatic, and anti-depressant properties. Moreover, vanillin exhibits neuroprotective effects on multiple neurological disorders and neuropathophysiological conditions. This study reviews the mechanisms of action by which vanillin prevents neuroinflammation and neurodegeneration in vitro and in vivo systems, in order to provide the latest views on the beneficial properties of this molecule in chronic neurodegenerative diseases and neuropathophysiological conditions.

## 1. Introduction

Vanillin is a natural phenolic aldehyde, extracted from the vanilla bean (Vanilla planifolia, Vanilla tahitensis, and Vanilla pompona) largely used as flavoring, usually in sweet foods and in the cosmetic industry [1] (Figure 1A). Besides the industrial use of vanillin, this natural compound possesses several beneficial effects for human health, mainly due to its strong antioxidant activity, in addition to its anti-inflammatory, anti-mutagenic, anti-metastatic, and anti-depressant properties [1,2,3]. Specifically, many studies have identified for vanillin a neuroprotective and anticancer effect, as well as antibiotic potentiation and anti-quorum sensing properties [1,3,4,5,6,7,8]. Moreover, vanillin is the main and the most stable degradation product of curcumin, a natural polyphenol with broad therapeutic benefits [9]. Nevertheless, curcumin is characterized by poor systemic bioavailability owing to its low solubility and chemical instability under physiological conditions. Recently, it has been proposed that the degradation products of curcumin, mostly including vanillin and ferulic acid, may make an important contribution to its biological activities [10,11]. Although recent studies on vanillin have suggested a strong bioactive potential for this molecule, the knowledge compared to curcumin is very limited.

Neuroinflammation is a common pathological process of many neurological diseases and it is also closely related to the underlying mechanisms of neurodegenerative disorders. Indeed, both neuroinflammation and oxidative stress are considered the main hallmarks of neurodegeneration. The anti-inflammatory effects of vanillin and its protective activity against oxidative damage are well established [2,3]. In particular, vanillin possesses a strong scavenger activity for ROS as observed in multiple antioxidant assays both in vitro and in vivo, and is able to inhibit inflammatory pathways activation [1,2,12,13]. In this respect, vanillin could be a promising therapeutic molecule to prevent neurological disorders. In addition, vanillin does not show toxic effects even at high concentrations, it is rapidly absorbed and, also, able to cross the blood–brain barrier, thus becoming a qualified candidate for the treatment of neurological diseases [14].

Given the data on the antioxidant and anti-inflammatory properties of vanillin, and considering its importance in its several applications and pharmacological properties, in this review we discuss the role of vanillin as a neuroprotective agent by analyzing its effects on several neurological disorders and in neuropathophysiological conditions including neuroinflammation.

## 2. Protective Effects of Vanillin on Neurological Diseases

### 2.1. Neurodegenerative Diseases

#### 2.1.1. Alzheimer’s Disease

Alzheimer’s Disease (AD) is an age-related neurodegenerative disease in which memory loss and cognitive decline are the prominent symptoms. The main neuropathological hallmarks of AD are the formation of neurofibrillary tangles (NFT) and the deposition of Aβ amyloid plaques in the brain. Moreover, cholinergic deficit as well as oxidative stress and neuroinflammation are involved in the processes leading to neuronal death [15,16,17]. The reduced activity of the neurotransmitter acetylcholine has been reported to be responsible for the cognitive decline associated with AD [18,19]. Currently, acetylcholinesterase (AChE) inhibitors are considered the main drugs for the treatment of AD [20,21]. In addition, diets rich in polyphenols have been shown to prevent the oxidative stress linked to the AD pathogenesis [22,23,24]. In this respect, vanillin could exert beneficial effects on AD due to its strong anti-inflammatory and anti-oxidant activity [2,3].

It has been shown that vanillin exhibits AChE inhibitory activity in vitro and in vivo, prevents amyloid aggregation, and also protects from neuronal oxidative damage. Ahmmad and coworkers have reported that vanillin showed a significant AChE inhibitory activity (IC_50_ 84.66 ± 3.20 µg/mL) as well as a strong anti-oxidant activity through free radical scavenging ability, reducing power, and lipid peroxidation reducing capacity [25]. AChE inhibitory activity (IC_50_ 5.6 µg/mL) of vanillin, was first reported by Kundu and Mitra [26]. Moreover, the inhibition of AChE activity as well as restore of oxidative imbalance by vanillin has also been reported in Fe^2+^-induced brain tissue damage [27].

In the last years, much attention has been paid to vanillin derivatives synthesis in order to improve the properties of the natural compound. Blaikie and coworkers have demonstrated that naphthalimide and phthalimide vanillin derivatives exhibited higher antioxidant potential and enhanced ability to inhibit cholinesterase in comparison to vanillin [28] (Figure 1B,C). Moreover, vanillin derivates showed selectivity for butyrylcholinesterase (BuChE), an isoform of AChE founded to increase its activity as the severity of the disease advances. In addition, the compounds also exhibited lipophilicity and blood–brain barrier permeation properties [28]. Similarly, vanillin derivatives, bearing a tacrine or a naphthalimide moiety, exhibited improved antioxidant properties and inhibitory activity toward AChE at micromolar concentrations compared to vanillin [29] (Figure 1D). In addition, these compounds inhibited Aβ amyloid aggregation and protected SH-SY5Y cells from H_2_O_2_-induced cell damage [29]. The AChE inhibitory activity of vanillin has been also observed in vivo, in the AlCl_3_ mouse model of AD. Long-term administration of AlCl_3_ to mice causes deficits in spatial reference, memory, and spatial working memory [30]. In this respect, vanillin administration with 30 mg/kg per day, preserved intact acetylcholine and significantly improved spatial reference memory. However, higher doses of vanillin (60 and 120 mg/kg per day) caused a significant depletion in total ACh levels. The observed benefits with lower doses of vanillin were attributed to vanillin’s ability to indirectly increase synaptic acetylcholine abundance through inhibiting AchE [30]. Moreover, a previous study indicated that vanillin was able to confer neuroprotection in an AlCl_3_ mouse model of AD acting as a selective agonist of donepezil, a potent AChE inhibitor [31].

A key neuropathological feature of AD is the deposition of Aβ amyloid plaques in the brain. Increased aggregation of Aβ peptides in neurotoxic oligomers promotes oxidative stress triggering a pathological cascade that eventually leads to cell death. Ethyl vanillin (EVA), an analogue of vanillin isolated from vanilla beans, has been found to protect PC12 cells from Aβ1-42 aggregate-induced oxidative injury by restoring the antioxidative enzymes’ activity and reducing intracellular lipid peroxidation. In addition, EVA was able to prevent cell viability reduction and Aβ-induced apoptosis by attenuating caspase-3 activation and increasing the Bcl-2/Bax ratio [32].

The overall data suggest that, due to the many contributing factors involved in the development and progression of AD, vanillin could be a potential multi-target compound for the treatment of this neurodegenerative disease.

#### 2.1.2. Parkinson’s Disease

Parkinson’s Disease (PD) is a progressive and age-related neurodegenerative disease mainly affecting motor function. The pathological hallmark of PD is the selective loss of dopaminergic neurons in the substantia nigra resulting in striatal dopamine depletion. Moreover, it is characterized by Lewy Bodies, proteinaceous cytoplasmic inclusions consisting mainly of α-synuclein aggregates. Although the precise pathophysiology underlying the development of PD is unknown, accumulating evidence suggests a potential role of mitochondrial dysfunction, oxidative stress, and neuroinflammation in the pathogenesis of PD [33,34,35]. Recently, growing evidence has accumulated demonstrating the beneficial role of vanillin in both in vitro and in vivo experimental models of PD.

Oxidative stress is believed to trigger a molecular pathway cascade responsible for inducing dopaminergic neuron degeneration [33]. Vanillin has been reported to show neuroprotective effects in SH-SY5Y neuroblastoma cells treated with rotenone (an in vitro model of PD) by its anti-oxidant and anti-apoptotic properties [36]. Rotenone is reported to lead to dopaminergic neuronal apoptosis through the activation of mitogen-activated protein kinase (MAPK) and caspase-dependent pathways [37,38]. Interestingly, the pre-treatment of neuroblastoma cells with vanillin was able to decrease p-JNK, p-P38, and p-ERK levels, cyt-C release, caspases 3, 8, and 9 activation, as well as down- and up-regulate Bax/Bcl-2, respectively. Moreover, vanillin prevented ROS production and ameliorated mitochondrial membrane potential alteration [36]. These observations have suggested that the underlying neuroprotective mechanism of vanillin in rotenone-induced apoptosis could be ascribed to its anti-oxidant action and its ability to preserve mitochondrial functions [36]. The protective role of vanillin in PD has also been tested using in vivo models. In particular, Dhanalakshmi and coworkers have investigated the neuroprotective role of vanillin by analyzing the motor deficits, the neurochemical variables, the oxidative and anti-oxidative indices, as well as the expression of apoptotic markers in a rotenone-induced rat model of PD [39]. Concerning the motor deficits, the Authors have reported that oral administration of vanillin was able to attenuate rotenone-induced behavioral impairments in a dose-dependent manner. Specifically, vanillin was able to improve the impairment of movement (peripheral and central) and the activities (grooming and rearing) in open field tests, as well as the akinetic and cataleptic impairment [39]. Moreover, vanillin (20 mg/kg) significantly reduced rotenone-induced depletion of striatal dopamine when co-administrated with rotenone. Regarding the effect of vanillin on oxidative stress induced by rotenone, vanillin was able to decrease lipid peroxidation, GSH levels and SOD, catalase, and GPx activity. Furthermore, rat treated with chronic rotenone showed significant induction in the expression of Caspases 3, 8, and 9 in substantia nigra and striatum, as well as an increase in pro-apoptotic factor Bax and simultaneous depletion of anti-apoptotic factor Bcl-2 expression. These alterations were significantly attenuated by co-treatment with vanillin when compared to rotenone alone-treated animals [36]. Recently, the protective effect of vanillin has also been investigated in a 6-hydroxydopamine (6-OHDA) rodent model of PD [40]. The rat 6-OHDA treatment for 14 days induced behavior alterations and a reduction in striatal dopamine concentration. Both oral and intraperitoneal vanillin administration at three days before or seven days after 6-OHDA treatment resulted in significantly increasing dopamine concentration and decreasing apomorphine-induced contralateral rotation. Although no mechanisms were investigated, vanillin seems to offer protective properties against 6-OHDA lesions via preserving striatal dopamine levels [40].

The progressive neurodegeneration in PD is also related to neuroinflammation [35,41,42,43]. A neuroprotective effect of vanillin, exerted by its anti-inflammatory action, has been reported by Yan and coworkers [44]. In this study, the Authors used lipopolysaccharide (LPS)-induced PD models, both in vivo and in vitro systems. In vivo, vanillin was able to improve motor dysfunction, suppress dopaminergic neuron degeneration, and inhibit microglial over-activation induced upon injection of LPS into the substantia nigra of PD rat models [44]. In vitro, vanillin was able to prevent the LPS-induced inflammatory response in murine microglial BV-2 cells. Vanillin markedly inhibited LPS-induced increase in the mRNA and protein levels of inducible nitric oxide (iNOS), cyclooxygenase-2 (COX-2), IL-1β, and IL-6 in a concentration-dependent manner by regulating ERK1/2, p38, and NF-kB signaling [44]. The overall data indicated that vanillin has a role in protecting dopaminergic neurons via inhibiting inflammatory response activation.

#### 2.1.3. Huntington’s Disease

Huntington’s Disease (HD) is an autosomal dominant chronic neurodegenerative disease, caused by the expansion of a CAG (cytosine, adenine, and guanine) repeat in the huntingtin gene that promotes huntingtin’s aggregation, leading to deposition of cytoplasmic and intranuclear inclusion bodies. The main HD hallmark is motor dysfunction as well as cognitive and psychiatric deficits [45,46]. There are only a few studies on the effect of vanillin in HD. At first, Gupta and Sharma tested the effect of vanillin, as a selective agonist of transient receptor potential vanilloid subtype 1 (TRPV1), in 3-nitropropionic acid (3-NPA)-induced HD in rats (in vivo model of HD) [47]. TRPV1 is a non-selective cation channel, which is mainly expressed in peripheral sensory neurons, involved in the physiological functions of the cannabinoid system that includes motor coordination, memory processing, control of appetite, pain modulation, and neuroprotection [48]. It has been reported that TRPV1 activation is beneficial in alleviating specific motor symptoms in HD [49]. Gupta and Sharma have reported that vanillin administration exerted a protective effect against (3-NPA)-induced HD. Specifically, vanillin significantly attenuated 3-NPA-induced weight loss, impaired locomotion, motor coordination, and learning memory as well as biochemical impairments, such as brain striatum oxidative stress and ameliorating lipid peroxidation [47].

### 2.2. Other Neurological Diseases

#### Spinal Cord Injury

Spinal cord injury (SCI) is a neuronal disorder that causes disability and paralysis and its pathogenesis has yet to be fully elucidated [50]. Several studies have suggested that the activation of mitochondria-mediated apoptosis plays an important role in the secondary damages induced by SCI [51,52]. Many proteins, including BcI-2-associated X (Bax), Bcl-2, mitofusin (Mfn)-l and -2, and dynamin-related protein 1 (Drpl), are involved in the changes that occur in mitochondria during apoptosis [53]. Additionally, hypoxia-inducible factor (HIF)- lα induces hypoxia-associated apoptosis by inhibiting the expression of Bcl-2 proteins. Bc12 and Bax are anti-apoptotic and apoptotic genes, respectively, which belong to the Bc1-2 family, and it is known that Ad-HIF-lα attenuates the altered expression of Bax/Bcl-2 ratio to protect against neuronal apoptosis [54]. Chen and co-workers have evaluated the effect of vanillin in a rat model of SCI [54]. Specifically, they have observed that the injection of 286 mg/kg of vanillin in rats reduces the levels of proinflammatory cytokines, restores the antioxidant activity of SOD and CAT, and reduces the levels of MDA. Interestingly, vanillin treatment promotes a reduction in apoptotic cells, attenuating the altered expression of mitochondrial proteins, BAX, CytC, and Bcl-2, associated with SCI. In addition, Chen and coworkers demonstrated that the observed effects of vanillin are mainly due to the ability of this natural compound to attenuate HIF-lα levels [54].

## 3. Protective Effects of Vanillin in Neuropathophysiological Conditions

### 3.1. Hypoxic-Ischemic Brain Injuries

Vanillin has been reported to exhibit neuroprotection against hypoxia-ischemia brain damage. Neonatal hypoxic-ischemic brain damage (HIBD) is the main cause of newborn deaths and unrecoverable and long-lasting neurodevelopmental disabilities in the sufferers. HIBD complications are estimated at around 23% of infant deaths all over the world and they affect over 1 million newborns per annum [55,56]. HIBD is caused by a restriction in blood supply (ischemia) and deprivation of adequate oxygen supply (hypoxia), which causes a switch to anaerobic metabolism. Upon reperfusion and reoxygenation, the renewed availability of oxygenated blood to the previously ischemic tissue causes the formation of additional ROS by mitochondria (via the electron transport chain) and damage the integrity of the membrane and organelle’s function. In addition, damaged neurons and activated endothelium produce various cytokines, including interleukins (IL-1, IL-6, IL-8, and IL-10) and tumor necrosis factor (TNF)-α, which activate inflammatory response [57]. Lan and coworkers have recently examined the neuroprotective effects of vanillin in an HIBD model of 7-day-old SD rats, administering 40 or 80 mg/kg of vanillin every 12 h for two consecutive days [58]. After this time, the authors have observed that the presence of vanillin promotes early neurofunction development, ameliorates histomorphological damage (in particular alleviates the damage of BBB ultrastructure), and protects neuronal damage in the cortex and hippocampal CA1 and CA3 regions after HIBD in neonatal rats. A previous study conducted by Kim and coworkers also demonstrated that vanillin is able to prevent hippocampal CA1 cell death after transient global ischemia in Mongolian gerbils [59]. In particular, pre- and post-insult vanillin showed a significant increase in neuron survival in treated animals and this effect has been ascribed to the ability of vanillin to inhibit oxidative stress and excitotoxicity as well as suppress neuronal death inCA1 region [59].

The hippocampus is especially susceptible to chronic intermittent hypoxia (CIH), which is the most crucial feature of obstructive sleep apnea (OSA). Toll-like receptor 2 (TLR2), mostly expressed in microglia in the brain, is pivotal for the development of numerous hippocampal diseases and its neurotoxicity is mediated by its interaction with its adapter protein myeloid differentiation factor 88 (MYD88). Deng and coworkers have evaluated the effect of vanillin in adult male mice subjected to 8 h of intermittent hypoxia per day for 28 days [60]. Vanillin is an inhibitor of TLR2 and this study shows that the presence of vanillin strongly reduces the inflammatory response by limiting the interaction between TLR2 and MYD88 responsible for the neuronal damage in OSA. In addition, this study shows that vanillin reduces the activation of NF-kB and, consequently, the production of TNF-α, IL-6, and IL1β, both in the hippocampus and serum of the animals. Moreover, also in this case, the antioxidant activity of vanillin strongly contributes to the observed protective effect [60].

### 3.2. Brain Toxins

Humans are both intentionally and accidentally exposed to a broad environmental xenobiotic and/or toxins, most of them able to induce oxidative stress that causes free radical generation [61,62]. The brain is particularly sensitive to toxic compounds, owing to the limitative regenerative capability of neurons. Therefore, the cellular toxicity of such compounds could potentially cause irreversible disruption of neuronal function. Among xenobiotics, potassium bromate (KBrO_3_) is particularly relevant for its toxicity and the effect of vanillin on neurotoxicity associated with KBrO_3_ has been investigated in adult mice by Saad and coworkers [63]. The Authors observe that the intraperitoneal injection of vanillin 100 mg/kg is able to improve all the parameters associated with KBrO_3_-induced oxidative stress. Specifically, vanillin prevents the alteration in brain fatty acid composition, and lipid peroxidation, and restores the activity of antioxidant enzymes altered by KBrO_3_. Strictly related to oxidative stress, KBrO_3_ also promotes inflammation, increasing the mRNA level of IL1β, IL6, TNF-α, and COX2 pro-inflammatory cytokines. Interestingly, upon cotreatment with vanillin, the levels of these mRNA are restored, thus indicating that vanillin also exerts anti-inflammatory activity. Furthermore, vanillin restores the activity of BuChE and AChE modified upon KBrO_3_ treatment [63].

Physiologically, iron (Fe^2+^) plays an important role in the normal functioning of the brain as it acts as a co-factor for most neurotransmitter enzymes [64]. However, high concentrations of Fe^2+^ have been implicated in the exacerbation of neurodegeneration via excessive generation of reactive oxygen species (ROS); in this case, iron is considered a brain toxin [65]. Salau and co-workers have evaluated the effect of vanillin in Fe^2+^-induced brain damage ex vivo [27]. Specifically, they tested different concentrations of vanillin (from 15 to 240 µg/mL) for evaluating its therapeutic effects on oxidative imbalance, cholinergic and nucleotide-hydrolyzing enzyme activity, and dysregulated metabolic pathways. The ex vivo experiments, conducted on rat brain tissue, showed that vanillin exerts its anti-oxidant activity, restoring the levels of GSH, SOD, and CAT activity, decreasing lipid peroxidation and NO levels, thus indicating that vanillin reverses the oxidative damages induced by Fe^2+^. Moreover, this study shows that vanillin decreases the activity of BuChE and AChE and increases that of ATPase and ENTPDase [27].

Similar results have been obtained by Makni and colleagues, who evaluated the effect of vanillin on oxidative stress induced by carbon tetrachloride (CCl_4_) [66]. This molecule is xenobiotic and causes intoxication in animals, and it is considered a suitable molecule to simulate oxidative stress in many pathophysiological conditions [67,68]. The authors have performed this study in Wister rats, pretreating them with vanillin 150 mg/kg and then treating them with CCl_4_. The results clearly show that vanillin, also in the oxidative stress induced by CCl_4_, restores the activity of GSH, SOD, and CAT antioxidant enzymes and reduces MDA and NO levels, compared to CCl_4_-treated rats.

Hyperglycemia is known to promote the non-enzymatic reaction between sugars and amino-containing compounds, leading to protein glycation and the formation of Advanced Glycation End-products (AGEs). It is widely accepted that a direct link between diabetes and AD as protein glycation has been shown to enhance the neurotoxicity of Aβ peptides and tau protein and promote their aggregation and accumulation in vivo [69,70,71]. Moreover, protein glycation is increased nearly twofold in the cerebrospinal fluid of AD patients, including elevated levels of glycated albumin, apolipoprotein E, and transthyretin [72]. For this reason, AGEs could be considered brain toxins in this pathological context. Iannuzzi and coworkers have evaluated the effect of vanillin in AGE-induced ROS production in SH-SY5Y neuronal cells, showing that vanillin exerts cytoprotective and anti-oxidant effects [73].

### 3.3. Microglial Activation—Neuroinflammation

Neuroinflammation is a common pathological process in several neurological diseases, and it is associated both with the mechanisms underlying neurodegenerative diseases, such as Alzheimer’s (AD) and Parkinson’s (PD) diseases, and infectious neuropathology [74]. In a chronic oxidative stress state, the increase in reactive species, such as ROS and nitrogen reactive species (RNS), induces alteration in the intracellular signaling pathways involved in cell metabolism, leading to an enhanced secretion of proinflammatory molecules, thus promoting a dysregulation of the inflammatory response. In the central nervous system (CNS), the main player in neuroinflammation is microglia, a type of differentiated tissue macrophage, responsible for immune responses and homeostasis of CNS. Activated microglial cells induce the production of proinflammatory cytokines and inflammatory mediators, including interleukin (IL)-1β, IL-6, nitric oxide (NO), and tumor necrosis factor (TNF)-α, which amplify inflammatory responses in the brain leading to neuroinflammation. Nuclear factor-kappa B (NF-kB) and mitogen-activated protein kinase (MAPK) are two critical signaling pathways regulating pro-inflammatory cytokines expression during the inflammation process [75,76].

Microglia, a type of differentiated tissue macrophage, are responsible for immune responses and the homeostasis of the central nervous system (CNS). Microglia play a complex role in neuroinflammation, which is implicated in neurodegenerative diseases such as Alzheimer’s disease (AD) and Parkinson’s disease (PD) [74]. In the previous sections, we have cited several studies that have investigated the effect of vanillin on neuroinflammation in PD and AD. Considering the pivotal role of microglia in the homeostasis of CNS, we have decided to discuss the effect of vanillin on microglia activation in a specific paragraph.

At first, Kim and coworkers investigated the mechanism by which vanillin induces anti-neuroinflammatory responses in LPS- stimulated BV-2 microglial cells [77]. They observed that vanillin significantly inhibits NO production in LPS-stimulated BV-2 microglial cells, through the suppression of both mRNA and protein levels of iNOS and decreasing COX2 only at the protein level, but it has no effect on COX2 mRNA. Moreover, vanillin is able to reduce NF-kB activation, thus diminishing the expression of proinflammatory cytokines (IL-1β, IL-6, and TNF-α) and MAPK phosphorylation induced by LPS in BV-2 microglial cells [77].

Moreover, as previously reported, vanillin has a role in protecting dopaminergic neurons through the inhibition of inflammatory response activation [44]. Indeed, vanillin is able to reduce the LPS-induced expression of iNOS, COX-2, IL-1β, and IL-6 in murine microglial BV-2 cells by regulating ERK1/2, p38, and NF-kB signaling [44].

More recently, Ullah and coworkers have reported that vanillin acid, an oxidized form of vanillin, is able to inhibit the LPS-induced glial cell activation in mice brain [78]. Specifically, the authors showed that the LPS-vanillic acid co-treatment suppressed p-JNK, p-NF-kB, and associated pro-inflammatory mediators (TNF-α, IL-1α, and NOS-2) in the cortex and hippocampus region of mouse brain. Based on these findings, it has been suggested that the anti-inflammatory effect of vanillic acid against LPS-induced neuroinflammation might be via the inhibition of the JNK-mediated p-NF-kB signaling pathway [78].

Microglia play a central role also in glioma promotion and growth and they contribute to the tumor tissue by up to 30% of the cells. Glioma induces the polarization of glioma-associated microglia into a glioma-supporting phenotype through complex interactions involving several signaling mechanisms [79,80]. The main mechanism that contributes to glioma invasiveness is the degradation of the surrounding extracellular matrix by matrix metalloproteases (MMPs), especially MMP14. While microglia in the healthy brain do not express MMP14, it is upregulated in glioma-associated microglia through TLR-2 [81]. Triller and coworkers have investigated the impact of vanillin, as a TLR-2 inhibitor, to block the TLR2-mediated microglia pro-tumorigenic phenotype using in vitro and in situ microglia–glioma interaction experimental models [82].

Specifically, in primary microglial cells, the authors observed that 100 µM vanillin is able to reduce the IL-6 protein levels and MMP9, MMP14, and iNOS RNA expression, induced by TLR-2, thus suggesting that vanillin is a highly efficient inhibitor of TLR2 in murine microglia. Moreover, these results have been confirmed in human glioma-associated microglia, isolated from human glioma samples, where vanillin is able to reduce the RNA expression of MMP14 in 26% of glioma-associated microglia, while the reduction in MMP9 is not significant.

Furthermore, Triller and coworkers have monitored glioma growth in organotypic brain slices in the presence of vanillin, and they show that the treatment with vanillin 100 µM significantly reduces the tumor volume by 85% compared to an untreated control [79]. They have also tested whether the impact of vanillin is mediated by microglia by depleting these cells from the slices. As previously described, there was a reduction in glioma growth through the depletion of microglia, but the treatment with vanillin even leads to a further decrease in glioma growth [83].

## 4. Conclusions

To date, vanillin has been used mainly as a flavor and fragrance agent. Nevertheless, vanillin possesses different bioactivities that can be employed for humans. The studies presented in this review reveal several beneficial properties of vanillin in neuroprotection. Although different molecular mechanisms are involved in the prevention of neuroinflammation and neurodegeneration by vanillin, positive effects have been observed in several neurodegenerative disorders and neuropathophysiological conditions, both in vitro and in vivo. Indeed, vanillin exerts its protective effect mainly through anti-oxidant and anti-inflammatory effects that are strictly interconnected and key mechanisms in neurological disorders (Figure 2). These observations suggest a potential application for vanillin in the treatment and prevention of these neurological disorders; clearly, translation to clinical trials is needed for the usage of this nutraceutical as co-adjuvant therapy. Furthermore, future studies could be focused on nanocarrier systems for vanillin that may increase its stability, bioavailability, and bioactivity. In conclusion, vanillin seems to be a promising, accessible, and novel neuroprotective agent and it could be used as functional food and a food supplement for the prevention of neurological disorders.

## Figures and Tables

**Figure 1 ijms-24-01817-f001:**
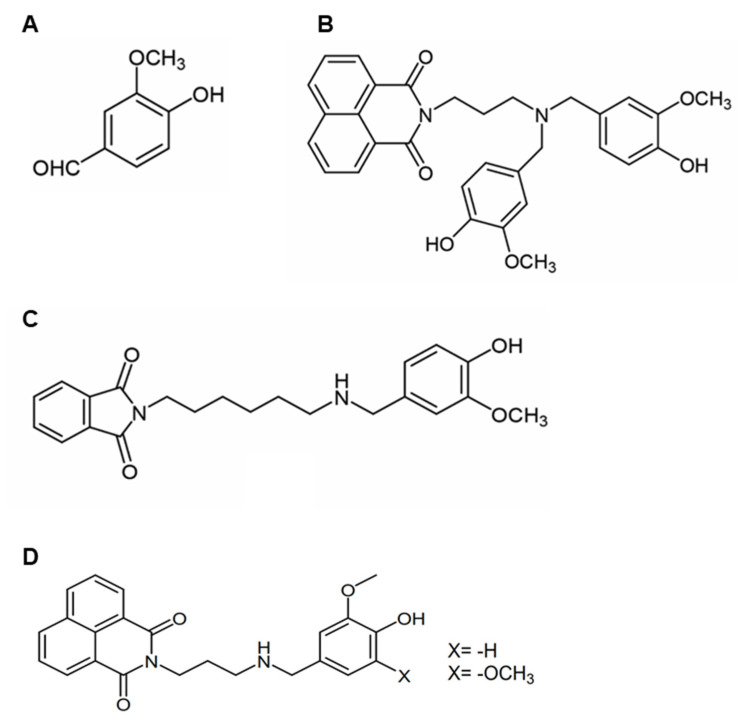
Chemical structure of vanillin and its derivatives: (**A**) Vanillin; (**B**) di-vanillin derivative with naphthalimide; (**C**) mono-vanillin derivative with phthalimide; (**D**) vanillin–naphthalimide compounds.

**Figure 2 ijms-24-01817-f002:**
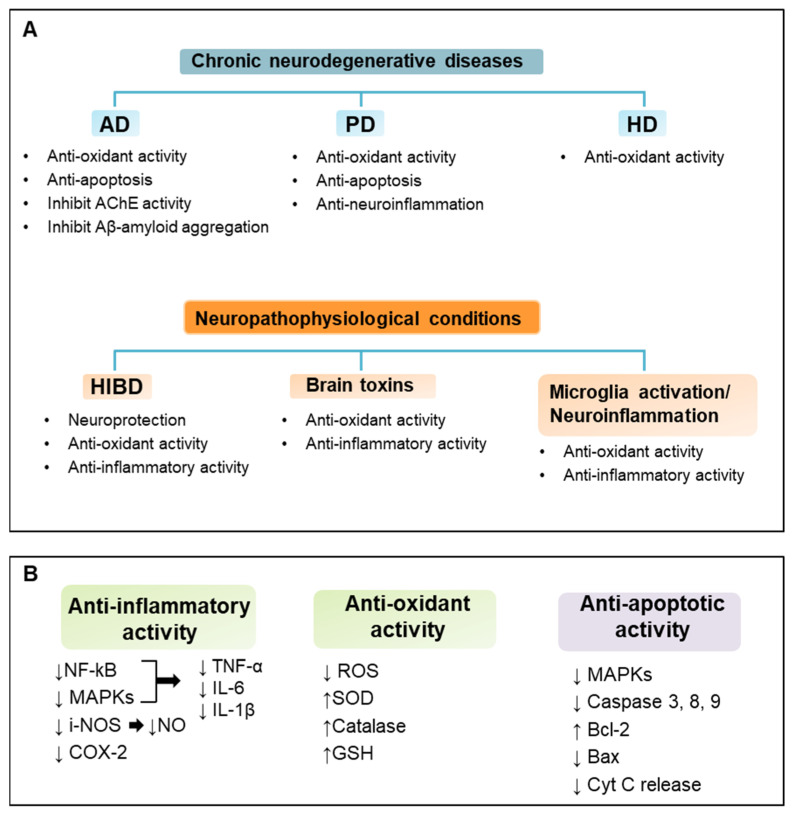
Neuroprotective effects and mechanisms of vanillin. Vanillin exerts multiple different protective effects on various neurological conditions. Main neuroprotective effects and mechanisms of vanillin on chronic neurodegenerative diseases and neuropathophysiological conditions are summarized in panel (**A**). Molecular signaling pathways involved in vanillin anti-inflammatory, anti-oxidant, and anti-apoptotic activity are outlined in panel (**B**).

## Data Availability

Not applicable.

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
