# Peer review of "Overview of the Role of Vanillin in Neurodegenerative Diseases and Neuropathophysiological Conditions"

_ijms, 2023, doi:10.3390/ijms24031817_

Round 1

Reviewer 1 Report

Originally, vanillin is natural product but most of the currently commercially available vanillin (>85%) are produced via synthetic routes (ChemistryOpen. 2019 Jun; 8(6): 660–667. doi: 10.1002/open.201900083). Vanillin and its derivatives show polypharmacologic bioactivities. The current manuscript #ijms-2110849 provides a minireview of vaniline and its derivatives for management of neurodegenerative disease but not for neurological diseases, which is a broader term that involves also psychiatric diseases and neurodevelopmental diseases. Therefore, the inaccurate title should be corrected. The manuscript is completely devoid from any illustrative figure which is a major drawback. Illustrative figures should be introduced. They make it easier for the reader to get the major facts and understand the text. In addition, the manuscript should presents chemical structures of compounds especially when it presents derivatives such as naphthalimide, phthalimide derivatives and others. In addition, it should differentiate between vanillin itself and its derivatives. I wonder why the manuscript is some parts stresses that vanillin extracted from a certain source shows some biological activity (e.g. lines 70 and 74). Cannot vanillin from other sources (natural or synthetic) show the same activity?

Author Response

Please consider the file attached.

Reviewer 2 Report

Iannuzzi et al aim here to address the role of vanillin as a neuroprotective phytochemical against pathophysiological conditions (including neuroinflammation) in several neurodegenerative diseases.

Although the subject is very interesting - specially by linking a typical spice with its healthy properties - I feel like the authors could collaborate more by presenting an improved and integrative comprehension of vanillin (and generally other phytochemicals) importance for human health. The authors would succeed that by presenting & discussing the molecular and cellular/signaling mechanisms involved in such neuroprotective properties. Therefore, I have MAJOR considerations that should be solved before this review`s acceptance in IJMS/MDPI:

(i) The whole structure of the review: from my point of view, the authors should extend the "neuroinflammation" section, mentioning anti- and pro-inflammatory cytokine dynamics (e.g. chronic effects) and include 2-3 paragraphs linking the metabolism of Reactive Oxygen/Nitrogen Species (ROS/RNS), oxidative stress, and neuroinflammation. 

After that, one additional section should be also included, approaching all cellular/signaling mechanisms triggered by vanillin that would result in antioxidant and anti-inflammatory properties. Authors should look for Keap1-Nrf2-AREs and NF-kB cascades, at least.

Please, include comprehensive figures/schemes for that.

(ii) most of the active phytochemicals and nutraceuticals activate those aforementioned cascades through Michael`s reaction on sensitory thiol (-SH) groups of redox signaling molecules in target cells, in this case, neurons, astrocytes, etc in nervous system. I presume that kind of information is valious for IJMS/MDPI audience here

(iii) Please, correct chemical symbols all over the text. Coefficients should be subscripted, and charges superscripted, e.g. lines 76, 87, 89, 253, 264, 283, etc.

(iv) please, standardize the IC or EC units all over the text. e.g. lines 71, 91, 93, 141, 256, etc. 

Author Response

Please consider the file attached.

Round 2

Reviewer 1 Report

Current revised version might be suitable.

Reviewer 2 Report

I here consider that the authors properly succeeded in responding my major concerns about their MS first version.

Therefore, I also consider this review ready for publication in IJMS/MDPI.